# INTERPRETING CLASS CONDITIONAL GANS WITH CHANNEL AWARENESS

## ABSTRACT

Understanding the mechanism of generative adversarial networks (GANs) helps us better use GANs for downstream applications. Existing efforts mainly target interpreting unconditional models, leaving it less explored how a conditional GAN learns to render images regarding various categories. This work fills in this gap by investigating how a class conditional generator unifies the synthesis of multiple classes. For this purpose, we dive into the widely used class-conditional batch normalization (CCBN), and observe that each feature channel is activated at varying degrees given different categorical embeddings. To describe such a phenomenon, we propose *channel awareness*, which quantitatively characterizes how a single channel contributes to the final synthesis. Extensive evaluations and analyses on the BigGAN model pre-trained on ImageNet reveal that only a subset of channels is primarily responsible for the generation of a particular category, similar categories (*e.g.*, cat and dog) usually get related to some same channels, and some channels turn out to share information across all classes. For good measure, our algorithm enables several novel applications with conditional GANs. Concretely, we achieve (1) versatile image editing via simply altering a single channel and manage to (2) harmoniously hybridize two different classes. We further verify that the proposed channel awareness shows promising potential in (3) segmenting the synthesized image and (4) evaluating the category-wise synthesis performance. Code will be made publicly available.

## 1 INTRODUCTION

The past few years have witnessed the rapid advancement of generative adversarial networks (GANs) in image synthesis (Karras et al., 2021; Brock et al., 2019). Despite the wide range of applications powered by GANs, like image-to-image translation (Isola et al., 2017), super-resolution (Chan et al., 2021; Menon et al., 2020), and image editing (Ling et al., 2021), it typically requires learning a separate model for a new task, which can be time and resources consuming. Some recent studies have confirmed that a well-trained GAN model naturally supports various downstream applications, benefiting from the rich knowledge learned in the training process (Bau et al., 2019; Shen et al., 2020). Therefore, to make sufficient use of a GAN, it becomes crucial to explore and further exploit its internal knowledge.

Many attempts have been made to understand the generation mechanism of GANs. It is revealed that, to produce a fair synthesis, the generator is required to render multi-level semantics, such as the overall attributes (*e.g.*, the gender of a face image) (Shen et al., 2020), the objects inside (*e.g.*, the bed in a bedroom image) (Bau et al., 2019; Yang et al., 2020), the part-whole organization (*e.g.*, the segmentation of the synthesis) (Zhang et al., 2021), *etc*. However, existing efforts mainly focus on interpreting unconditional GANs, leaving conditional generation as a black box.

Compared with unconditional models, a class conditional model is more informative and efficient in that it unifies the synthesis of multiple categories, like animals, vehicles, and scenes (Brock et al., 2019). Figuring out how it manages the class information owns much great potential yet rarely explored. To fill in this gap, we take a close look at the popular class-conditional batch normalization (CCBN) (Brock et al., 2019), which is one of the core modules distinguishing conditional generators from unconditional ones. Concretely, CCBN learns category-specific parameters to scale and shift the input features, such that the output features developed with different class embeddings can be

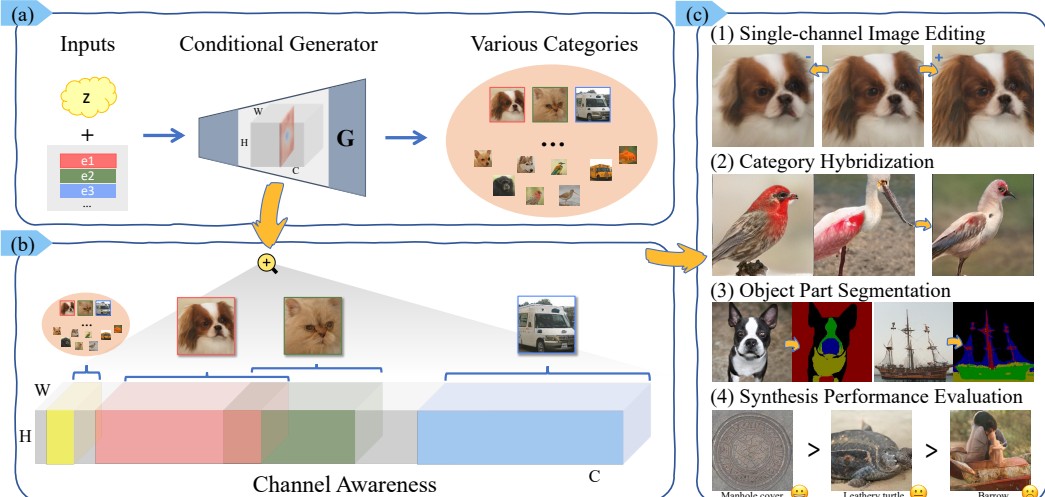

Figure 1: **Novel applications enabled by interpreting class conditional GANs.** Given a conditional generator in (a), we propose *channel awareness* to quantify the contribution of each feature channel to the output image, as shown in (b), which reveals how the categorical information is handled by different channels. **Red**, **green**, and **blue** channels are primarily responsible for the synthesis of a particular category, while **yellow** ones are shared by all classes. (c) Such an interpretation facilitates a range of applications, including single-channel image editing, category hybridization, fine-grained semantic segmentation, and category-wise synthesis performance evaluation. (Zoom in for better view.)

easily told apart from each other, eventually resulting in the synthesis of various categories. We notice from such a process that, under the perspective of the ReLU activation (Nair & Hinton, 2010) following CCBN, different feature channels present varying behaviors given different embeddings.

To quantify the aforementioned channel effect, we propose *channel awareness* that characterizes how a single channel contributes to the final synthesis. Through in-depth analyses on the BigGAN (Brock et al., 2019) model pre-trained on ImageNet (Deng et al., 2009), we have the following *key findings*, which are also illustrated in Fig. 1b. First, only a portion of channels are active in rendering images for a particular class while the remaining channels barely affect the generation. Second, more similar categories tend to share more relevant channels. For instance, channels regarding dog synthesis intersect with those of cats but disjoint from those of buses. Third, some channels highly response to the latent code instead of the class embedding and hence appear to deliver knowledge to all classes.

Beyond model interpretation, our proposed channel awareness facilitates a range of *novel applications with class conditional GANs*, as shown in Fig. 1c. First, after identifying the relevant channels through awareness ranking, we realize versatile image editing by simply altering a *single* feature channel (Sec. 5.1). Second, through mixing the channels that are related to two classes respectively, we achieve harmonious category hybridization (Sec. 5.2). Third, we verify that intermediate feature maps from the generator, after weighted by our channel awareness, can be convincingly used for fine-grained semantic segmentation (Sec. 5.3). Fourth, we empirically demonstrate the potential of our channel awareness in evaluating the category-wise synthesis performance (Sec. 5.4).

## 2 RELATED WORK

Among various types of generative models, such as variational auto-encoder (VAEs) (Kingma & Welling, 2013; Razavi et al., 2019), flow-based model (Kingma & Dhariwal, 2018), diffusion model (Ho et al., 2020; Dhariwal & Nichol, 2021), *etc.*, GAN (Goodfellow et al., 2014) has received wide attention due to its impressive performance on both unconditional synthesis (Karras et al., 2019; 2020; 2021) and conditional synthesis (Zhang et al., 2019; Brock et al., 2019; Sauer et al., 2022). Early studies on interpreting GANs (Bau et al., 2019; Shen et al., 2020) suggest that, a well-learned GAN generator has encoded rich knowledge that can be promising applied to various

downstream tasks, including attribute editing (Zhu et al., 2022; Yang et al., 2020; Ling et al., 2021), image processing (Pan et al., 2020; Zhu et al., 2020), super-resolution (Menon et al., 2020; Chan et al., 2021), image classification (Xu et al., 2021), semantic segmentation (Zhang et al., 2021; Xu & Zheng, 2021), and visual alignment (Peebles et al., 2022). Existing interpretation approaches usually focus on the relationship between the latent space and the image space (Shen et al., 2020; Zhu et al., 2021; Yang et al., 2020; Wu et al., 2021), hence commonly evaluated on unconditional models. Some attempts are made to also analyze class conditional models (Jahanian et al., 2020; Voynov & Babenko, 2020; Härkönen et al., 2020; Shen & Zhou, 2021), but they still target the latent space, leaving it unclear how the generator leverages the categorical information. This work clearly **differs** from prior arts from the following aspects. (1) We inspect the conditional generator from the channel perspective, which aggregates the messages from *both the latent code and the class embedding*. To our knowledge, this is the first attempt on understanding the function of embedding space in conditional generation. (2) We demonstrate the editability of altering a *single channel* of the conditional generator. Different from the single-channel editing in unconditional GANs (Wu et al., 2021), our approach identifies different relevant channels with respect to different categories in an *unsupervised* manner. (3) We achieve fine-grained semantic segmentation by paying more attention to some "important" channels. Unlike the existing efforts on single-object generation (Zhang et al., 2021; Xu & Zheng, 2021), our method does *not* require data-driven learning and can be robustly generalized to all classes. (4) We also enable some applications that are *peculiar to conditional models*, including the category hybridization and the category-wise synthesis performance evaluation.

## 3 METHODOLOGY

In this section, we introduce the proposed channel awareness. Specifically, we re-examine the class-conditional batch normalization (CCBN), which is widely used in class conditional generation, and investigate how it helps the generator with the categorical information provided by the class embedding. It is noteworthy that our approach is *fully unsupervised*, without relying on any segmentation masks or annotations.

### 3.1 PRELIMINARIES

Unlike unconditional GANs, where the generator takes the latent code $\mathbf{z}$, as the only input, a class conditional generator employs an additional embedding vector, $\mathbf{e}$, to provide the categorical information. Accordingly, the generation process can be formulated as $\mathbf{I} = G(\mathbf{z}, \mathbf{e})$, where $\mathbf{I}$ and $G(\cdot, \cdot)$ are the output image and the generator, respectively. That way, given a different embedding, the generator is able to produce images for that specific category.

There are many ways of integrating $\mathbf{e}$ into $G(\cdot, \cdot)$, where the most popular one is to adopt the class-conditional batch normalization (CCBN) (Brock et al., 2019). In particular, CCBN learns class-specific parameters to scale and shift the input feature maps as

$$\mathbf{y} = \boldsymbol{\gamma}(\texttt{concat}(\mathbf{z}, \mathbf{e})) \odot \frac{\mathbf{x} - \mu(\mathbf{x})}{\sigma(\mathbf{x})} + \boldsymbol{\beta}(\texttt{concat}(\mathbf{z}, \mathbf{e})), \tag{1}$$

where $\mathbf{x}$ and $\mathbf{y}$, both with shape $C \times H \times W$, denote the input and output features. $\mu(\cdot)$ and $\sigma(\cdot)$ compute the mean and variance of a tensor along the spatial dimensions (*i.e.*, $H$ and $W$). $\texttt{concat}(\cdot, \cdot)$ stands for the concatenation operation. $\boldsymbol{\gamma}(\cdot)$ and $\boldsymbol{\beta}(\cdot)$ outputs the $C$-dimensional scale and bias by learning from both $\mathbf{z}$ and $\mathbf{e}$. $\odot$ represents the element-wise multiplication with broadcasting.

### 3.2 CHANNEL AWARENESS

**Channel Probe.** From Eq. (1), we can tell that both the latent code and the class embedding act on the generation through CCBN. In other words, their messages are delivered to the feature channels through the learning of $\boldsymbol{\gamma}(\cdot)$ and $\boldsymbol{\beta}(\cdot)$. Now, we take a look at how each single channel contributes to the synthesis. For a particular channel with index $c$, Eq. (1) can be simplified as

$$\mathbf{y}^c = \gamma_e^c \mathbf{x}^c + \beta_e^c, \tag{2}$$

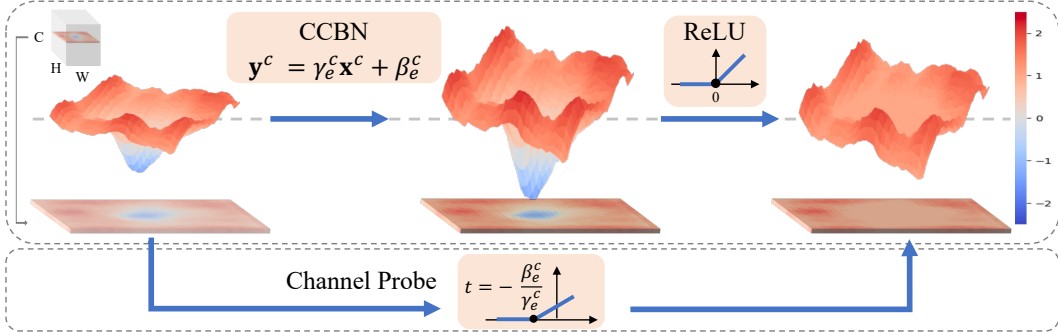

Figure 2: **Concept diagram of channel probe**, from whose statistics we develop the *channel awareness*. For a particular feature channel, it is first transformed by class-conditional batch normalization (CCBN) with category-specific scale $\gamma_e^c$, and bias $\beta_e^c$, then activated by ReLU to filter out negative neurons. Such a process is reformulated as a combined operation (as shown at the bottom), where we define the channel probe as $t = -\frac{\beta_e^c}{\gamma_e^c}$. This value reflects the integrated action of CCBN and ReLU.

where $\mathbf{x}^c$ and $\mathbf{y}^c$, both with shape $1 \times H \times W$, denote the normalized input (*i.e.*, subtracting mean and dividing by standard deviation) and the output. $\gamma_e^c$ and $\beta_e^c$ are scalars for the $c$-th channel. Here, $e$ stands for the embedding index, carrying the categorical information, and the effect of $\mathbf{z}$ is omitted for simplicity.

CCBN is usually followed by ReLU (Brock et al., 2019). Such an activation controls the information flow in that negative values in $\mathbf{y}^c$ are cut off to zero. According to Eq. (2), it is equivalent to cutting off the values in $\mathbf{x}^c$ that are smaller than

$$t_e^c = -\frac{\beta_e^c}{\gamma_e^c}. \tag{3}$$

In this way, we manage to directly relate the output feature channel to its corresponding input channel, as shown in Fig. 2. The value $t_e^c$ acts like a *channel probe* since it measures the channel-wise activation of a particular synthesis. Generally, for a certain channel, it presents different behaviours (*i.e.*, with different $t_e^c$ values) for different samples and different categories.

**Category-oriented Channel Awareness.** As discussed above, given a well-trained generator with CCBN, $t_e^c$ is strictly determined by the latent code, $\mathbf{z}$, and the class embedding, $\mathbf{e}$. Recall that the channel probe $t_e^c$ is instance-aware, which fluctuates along with the synthesis varying. To get a more reliable understanding of the function of a single feature channel, we derive *channel awareness* from the statistics of $t_e^c$. Oriented to the class embedding, $\mathbf{e}$, we would like to eliminate the impacts caused by the randomness of $\mathbf{z}$. To this end, we first sample a number of latent codes with the embedding fixed, then calculate the $t_e^c$ value for each synthesis and perform averaging. The resulting averaged score can be used to evaluate how a feature channel is responsible for the synthesis of a particular category. Recall that a lower $t_e^c$ suggests that more information will be preserved along this channel. Therefore, channels with lower mean values are more likely to be used for generating the target class. We define category-oriented channel awareness as $-\mathbb{E}_z[t_e^c]$.

**Latent-oriented Channel Awareness.** We further study the contribution of the latent code, $\mathbf{z}$, to each channel. Similarly, we sample a collection of latent codes with a fixed embedding and calculate $t_e^c$. Differently, this time we are interested in the variance instead of the mean, which gives us the latent-oriented channel awareness. A larger variance indicates that the randomness has a stronger influence on the synthesis regarding this channel. Hence, the latent-oriented channel awareness, $\text{Var}(t_e^c)$, reflects how sensitively a channel reacts to the latent code with the given embedding.

## 4 EVALUATION AND ANALYSIS

### 4.1 CATEGORY-ORIENTED CHANNEL AWARENESS

**Qualitative Evaluation.** We measure the causal channel effect on image generation via masking the target channels to zero during the forward process (Bau et al., 2019). Then we compare the

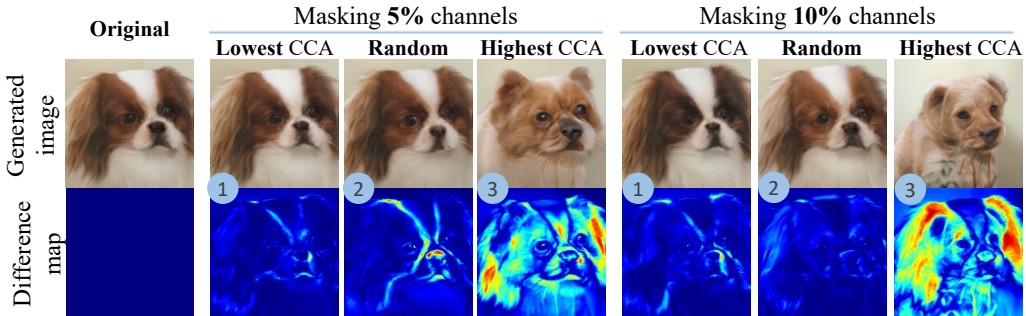

Figure 3: **Qualitative results** of masking channels with the lowest (the 1st column) and the highest (the 3rd column) category-oriented channel awareness (CCA) as well as masking randomly selected channels (the 2nd column, as baseline). Masking channels with the highest CCA results in the most noticeable class-relevant features lost, while masking channels with the lowest CCA causes only mild changes in the outputs. Results of more classes are shown in Appendix D.

generation results between masking channels with the highest and lowest channel awareness, and masking random channels, which is served as a baseline experiment. The qualitative results are shown in Fig. 3 which demonstrate that channels are not equally contributing to the generation of a particular class: Only partial channels (with the highest class awareness) primarily contribute to it, and some channels barely affect its generation (with the lowest class awareness).

**Quantitative Evaluation.** We further quantitatively verify the category-oriented channel awareness regarding every single channel in all 1,000 classes. We utilize a pre-trained classifier as an channel effect evaluator by assessing generated class images with and without a certain channel. Specifically, we measure the classification score drop after masking the target channel, which we refer to as categorical score change. We then estimate the correlation coefficient between the categorical score change and the proposed category-oriented channel awareness. We show four classes with the highest correlation in Fig. 4. For all 1,000 classes, the averaged correlation is 0.503 for BigGAN and 0.603 for BigGAN-Deep respectively. Details of evaluation on BigGAN-deep can be found in Appendix B. Note that a high correlation indicates the effectiveness of our method.

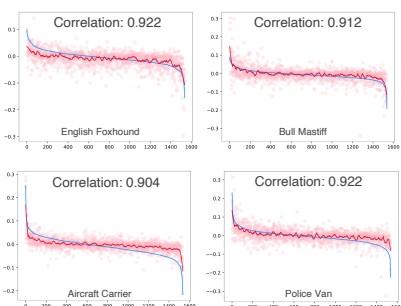

Figure 4: **Quantitative results** of the correlation between category-oriented channel awareness (blue line) and categorical score change (red line) of four classes. Top: The correlation between the two curves; Bottom: the corresponding class name. (The channel indices are sorted by the awareness scores for better display).

**Implementation details.** We exploit the representative BigGAN (Brock et al., 2019) and BigGAN-deep conditionally trained on the large-scale ImageNet (Deng et al., 2009) as the target models, which can generate realistic images of 1,000 various categories at $256 \times 256$ resolution. We use the Inception-v3 (Szegedy et al., 2016) model trained on ImageNet as the evaluation classifier, which is a widely used model for assessing generated images (Heusel et al., 2017). Then we calculate each channel's categorical score change before and after masking and take the average among 1,000 generated samples. The sampling process is without truncation for a fair estimation. To reduce data noise, we smooth the raw data of the categorical score change (see those pink dots in Fig. 4) via the Savitzky-Golay filter (Luo et al., 2005) with a window length 51. The smoothed curves are shown in red line in Fig. 4. The channel awareness (including both category-oriented and latent-oriented) is computed by sampling 10,000 samples per class, and then taking the average (for category) and variance (for latent) among samples of the per-channel channel probe scores. Please note that, the awareness scores are fixed after computation on a target generator, and can be directly used for the following analysis and applications.

**Channel overlap estimation.** After discovering high class-related channels, we wonder how distinct those channels are regarding different classes? Therefore, we estimate the channel overlap of two classes as $\frac{|S_i \cap S_j|}{k}$, where $S_i$ and $S_j$ represent the set of channel indices with top k highest awareness of class $i$ and $j$, and k is the number of selected channels. Results of channel overlaps

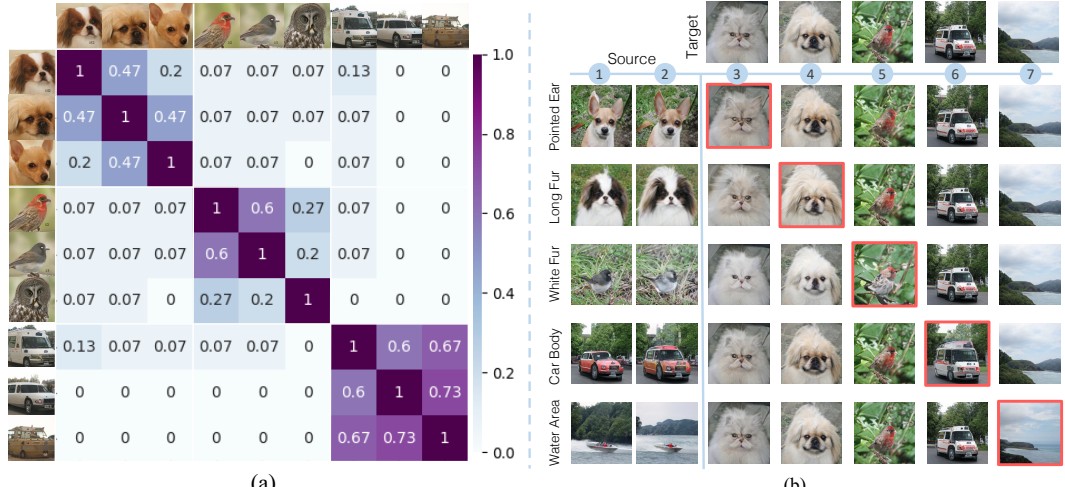

Figure 5: **Cross-class analysis on category-oriented channel awareness.** **(a)** Ratio of overlapped channels between 9 different classes. e select the top 100 channels with the highest category-oriented channel awareness for each class. A high ratio indicates more shared channels between the two classes. **(b)** Single-channel manipulation results on different classes. The semantics discovered from the source class (1st and 2nd columns) are applied to other target classes (3rd - 7th columns). (Zoom in for better view.)

Table 1: **Cross-class analysis on latent-oriented channel awareness.** Results of the number of shared channels among 1,000 classes in different layers (layer 0 - layer 11) in three types of channel selection criterion. For each class, we select only 10 channels (k=10) and perform intersection among all classes. The *category-oriented channel awareness* is served as baseline. Channel overlapping results of more ks, as well as an extreme case of k=1 can be found in Appendix C.

| Selection Criterion | L0 | L1 | L2 | L3 | L4 | L5 | L6 | L7 | L8 | L9 | L10 | L11 |
|---|---|---|---|---|---|---|---|---|---|---|---|---|
| highest awareness *w.r.t.* category | 0 | 0 | 0 | 0 | 0 | 0 | 0 | 0 | 0 | 0 | 0 | 0 |
| lowest awareness *w.r.t.* category | 0 | 0 | 0 | 0 | 0 | 0 | 0 | 0 | 0 | 0 | 0 | 0 |
| **highest awareness *w.r.t.* latent** | **3** | **3** | **3** | **1** | **3** | **3** | **0** | **4** | **2** | **1** | **5** | **4** |

of 81 class pairs and k = 100 are shown in Fig. 5a. This matrix indicates that visually similar classes(e.g., two different kinds of dogs) share more related channels, while distinct classes (e.g., dogs with cars) have nearly zero channel overlap.

**Single-channel manipulation.** Furthermore, we discover that channels with the highest category-oriented channel awareness control meaningful semantic information for that class, thus simply manipulating one target channel can achieve semantic editing. We provide the single-channel manipulation results in Fig. 5b, where each manipulation is achieved by multiplying a scale factor (=3) on the target channel during the forward process of generator. However, channels discovered by the source class can only manipulate itself as well as similar classes, while hard to manipulate other classes which do not have such attributes, as shown in Fig. 5b.

## 4.2 LATENT-ORIENTED CHANNEL AWARENESS

**Discovery of class-shared channels.** Different from the category-oriented channel awareness, which is used for finding highly class-related channels, the latent-oriented channel awareness is used for finding class-shared channels (even shared among all the 1,000 classes). Results in Tab. 1 demonstrate that, when only selecting 10 channels for each class of the highest latent-oriented channel awareness, then performing intersection on the sets of channel indices among all the classes, there still exists many channels left (the 3rd row of Tab. 1), while other two baseline experiments both have no overlapped channels (the 1st and 2nd row of Tab. 1). This implies that a class-conditional generator can learn unified representations provided to all classes. Besides, these class-shared channels can further enable semantic manipulation with highly similar effect on disparate classes. Visual editing results with respect to these channels are provided in Fig. 6b.

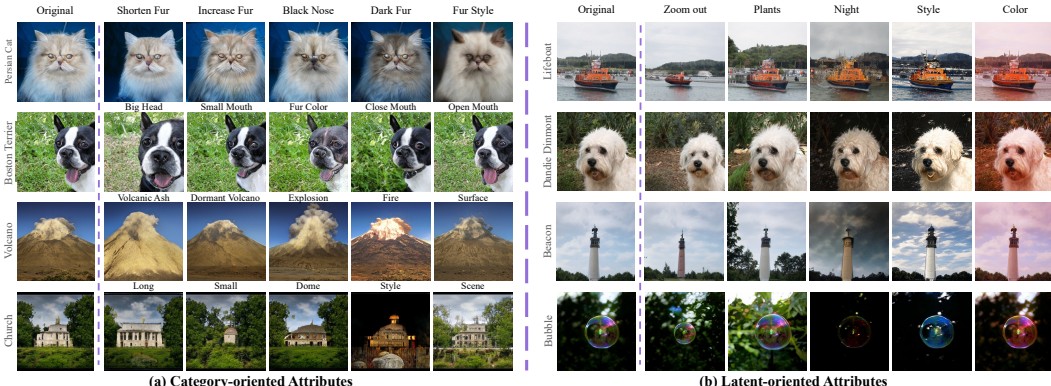

**(a) Category-oriented Attributes**         **(b) Latent-oriented Attributes**

Figure 6: **Versatile attributes discovered by the channel awareness. (a)** Attributes discovered via the category-oriented channel awareness, which are highly relevant to a particular class. **(b)** Attributes discovered via the latent-oriented channel awareness, which are shared among various classes. Each editing result is achieved by manipulating a single channel. (Zoom in for better view.) More results can be found in Appendix E and Appendix F.

## 5 APPLICATIONS WITH CLASS CONDITIONAL GANs

### 5.1 IMAGE EDITING VIA SINGLE CHANNEL MANIPULATION

With our method, we can identify meaningful channels through awareness ranking, from which we discover that simply manipulate a single channel can achieve versatile effects for image attribute editing. Such a manipulation can be either multiplying or adding a constant of manipulation magnitude. Therefore, with channels that are relevant to a particular class, we can discover and then manipulate various attributes of this class, such as fur, mouth, and nose for dogs and cats, explosion for volcanic. Results of image editing about **category-oriented attributes** are shown in Fig. 6a. Meanwhile, with channels that respond to latent, we can edit meaningful class-shared semantics controlled by the latent vector, which can therefore manipulate all kinds of classes in BigGAN. Results of image editing about **latent-oriented attributes** are shown in Fig. 6b.

### 5.2 CATEGORY HYBRIDIZATION VIA CHANNEL MIXING

In this section, we define a novel image editing task called *category hybridization*, which aims to synthesize realistic images while preserving meaningful attributes from more than one categories. This task could benefit novel content creation and produce unnatural yet plausible combinations of two categories which do not exist in the training set. To achieve this, we transplant the channels of the reference class to the corresponding channel positions in the feature of the input class, which we refer to as *channel mixing*. To activate the transplanting channels, we exploit the class embedding of the reference class for the generation of next layers. Results of category hybridization are shown in Fig. 7. With channel mixing, we could harmoniously fuse attributes from two categories into one realistic image which cannot achieve by style mixing (*i.e.*, simply mixes layer-wise class embeddings).

### 5.3 OBJECT PART SEGMENTATION VIA AWARENESS-WEIGHTED PIXEL CLUSTERING

Existing works (Xu & Zheng, 2021; Zhang et al., 2021; Abdal et al., 2021; Li et al., 2022; Tritrong et al., 2021; Ling et al., 2021) revealed that internal features in GANs can enable object part segmentation of generated images, which facilitates semantic annotation synthesis and local semantic editing tasks. Here, we present a simple unsupervised approaches for object part segmentation with the proposed channel awareness. Specifically, we collect the internal feature activations from multiple layers of the generator, then upsample each feature to be the same resolution as the target image $H' \times W'$. After upsampling, features from multiple layers could be concatenated along the channel dimension into a total feature volume as $C' \times H' \times W'$, where $C'$ is the total number of channels. A pixel-wise feature vector in the volume, in size of $C' \times 1 \times 1$, is considered as one sample for performing the clustering. Then, we simply perform $K$-Means

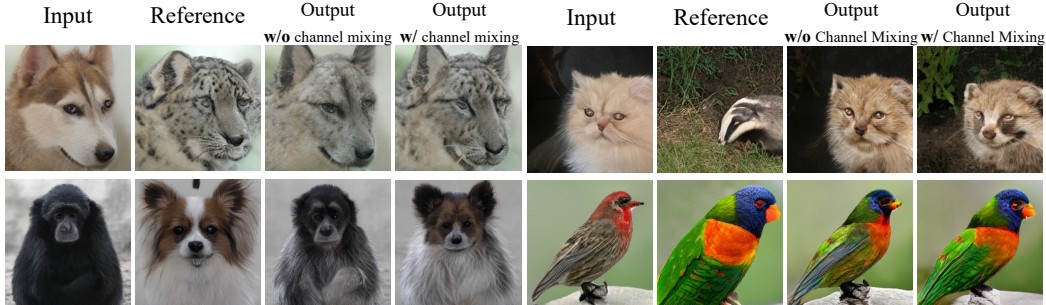

Figure 7: **Category hybridization by mixing the channels that are relevant to two categories.** Compare with the conventional style mixing (Karras et al., 2019) (the 3$^{rd}$ column), our approach (the 4$^{th}$ column) better fuses the characteristics (including both shape and appearance) of both classes. More results can be found in Appendix G.

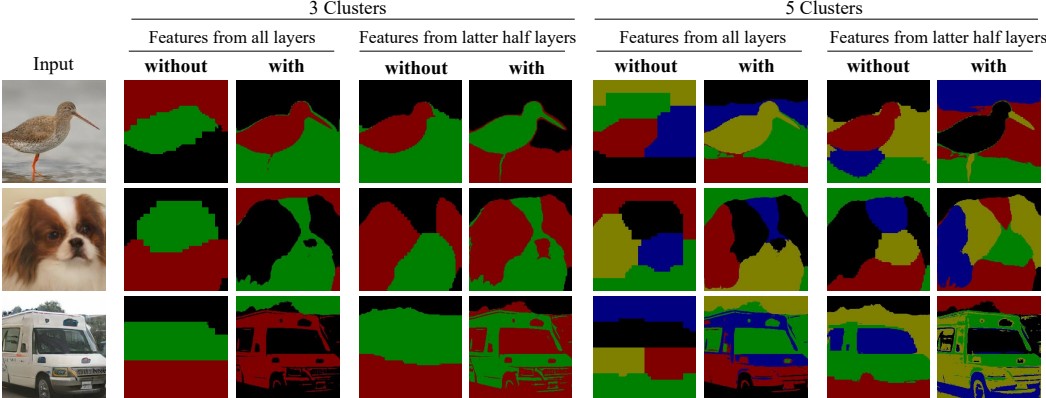

Figure 8: **Object part segmentation with and without awareness weighting**. We showcase that simple clustering on the feature space of the class-conditional GAN can obtain object part segmentations. Our channel awareness can be used for weighting channels and push them to be more class-related. Results of three different categories, two kinds of feature selecting layers, and two different amounts of clusters reveal that, in all settings, the channel awareness helps the clustering to **focus more on the class-relevant representations**, hence it recovers more detailed segmentations.

clustering on all pixel-wise feature vectors to segment the image into $K$ parts. Segmentation results without and with channel awareness are shown in Fig. 8.

## 5.4 SYNTHESIS PERFORMANCE EVALUATION VIA TOTAL CHANNEL AWARENESS

**Assumption and method.** For synthesis performance evaluation, our key assumption is that if one class has the *awareness* of consistently activating certain channels for producing category-oriented attributes while deactivating certain channels for suppressing irrelevant attributes, then the category will provide faithful generation results with high probability. For example, generating a flower needs to suppress those channels responsible for eyes and nose (mainly used by other classes like animals) and activate those related to petals. Thus, we measure the overall selection and suppression awareness as total channel awareness, by summing the absolute values of category-oriented channel awareness over all layers, which can be denoted as follows: $a_e = \sum_{c \in C'} |\mathbb{E}_z[t_e^c]|$. Here $C'$ stands for the number of total channels among all layers and $e$ is the given class.

**Results.** With total channel awareness, we empirically find that classes with the higher total channel awareness exhibit better quality yet low diversity of generated samples. Qualitative results of generated samples of classes with the highest and lowest total channel awareness are shown in Fig. 9. For quantitative verification, we calculate the correlation between total channel awareness and other four widely used metrics for GANs among all 1k classes. Specifically, we estimate the quality via precision (Kynkäänniemi et al., 2019), diversity via recall (Kynkäänniemi et al., 2019)

5 classes with highest total channel awareness

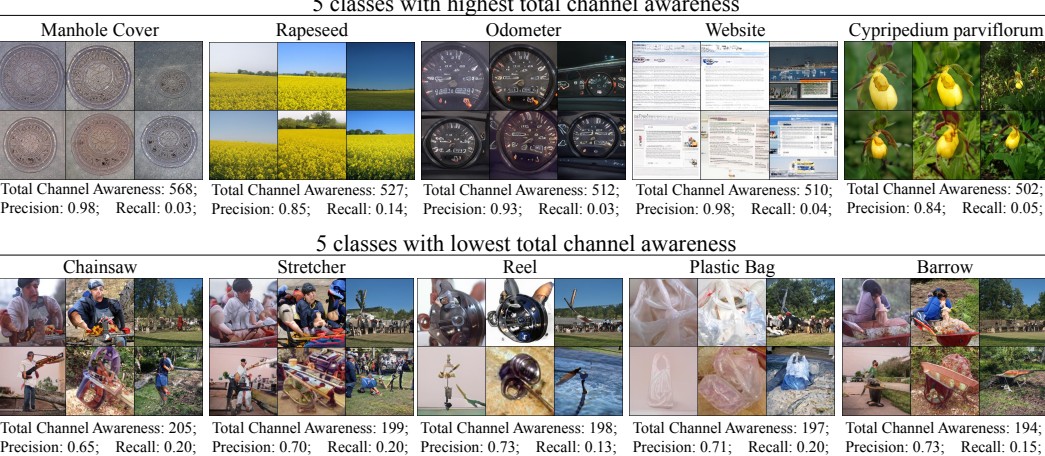

| Manhole Cover | Rapeseed | Odometer | Website | Cypripedium parviflorum |
|---|---|---|---|---|
| Total Channel Awareness: 568; | Total Channel Awareness: 527; | Total Channel Awareness: 512; | Total Channel Awareness: 510; | Total Channel Awareness: 502; |
| Precision: 0.98;  Recall: 0.03; | Precision: 0.85;  Recall: 0.14; | Precision: 0.93;  Recall: 0.03; | Precision: 0.98;  Recall: 0.04; | Precision: 0.84;  Recall: 0.05; |

5 classes with lowest total channel awareness

| Chainsaw | Stretcher | Reel | Plastic Bag | Barrow |
|---|---|---|---|---|
| Total Channel Awareness: 205; | Total Channel Awareness: 199; | Total Channel Awareness: 198; | Total Channel Awareness: 197; | Total Channel Awareness: 194; |
| Precision: 0.65;  Recall: 0.20; | Precision: 0.70;  Recall: 0.20; | Precision: 0.73;  Recall: 0.13; | Precision: 0.71;  Recall: 0.20; | Precision: 0.73;  Recall: 0.15; |

Figure 9: **Qualitative results of evaluating category-wise synthesis performance**, which is enabled by the *total channel awareness*. For each class, we show 6 generated samples, as well as total channel awareness, precision, and recall on below. We observe that classes with high total awareness tend to have high synthesis quality yet low diversity.

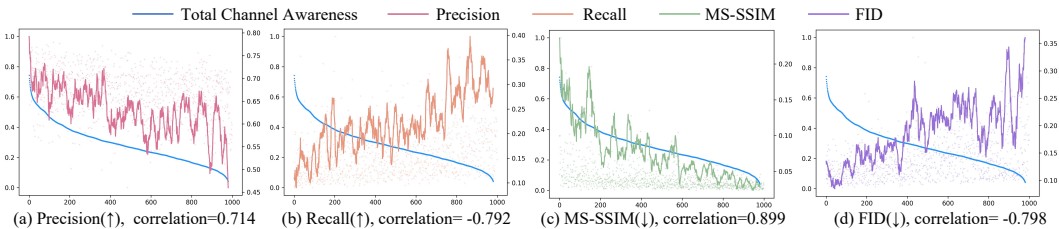

(a) Precision(↑),  correlation=0.714    (b) Recall(↑),  correlation= -0.792    (c) MS-SSIM(↓), correlation=0.899    (d) FID(↓), correlation= -0.798

Figure 10: **Quantitative results of evaluating category-wise synthesis performance**, including the correlation between our *total channel awareness* and the category-wise precision, recall, MS-SSIM, and FID. *X-axis:* Class indices sorted by the total awareness. *Y-axis:* Normalized total channel awareness and other four metrics. The correlations between total channel awareness and each other metric are shown at the bottom. ↑: higher is better; ↓: lower is better. Note that precision evaluates quality, recall and MS-SSIM evaluates diversity and FID measures both quality and diversity.

and MS-SSIM (Odena et al., 2017), and a general metric FID (Heusel et al., 2017) for each class. Implementation details are in Appendix H. The quantitative results are shown in Fig. 10.

## 6  CONCLUSION AND DISCUSSION

This work takes the first step towards understanding the generation mechanism of conditional GANs from the channel perspective. Concretely, we propose simple yet effective *channel awareness*, by which we successfully identify that channels that response to the latent code and to different class embedding. Extensive analyses shed light on how a conditional GAN manages the categorical information with different channels. More importantly, our channel awareness enables four novel applications with class conditional generators, which are rarely explored by prior work. In particular, we achieve single-channel attribute editing as well as harmonious category hybridization in a fully unsupervised manner. We also demonstrate the promising potential of our channel awareness in fine-grained semantic segmentation and category-wise synthesis performance evaluation.

Despite the appealing results, there are still some directions worth exploring. Recall that this work primarily targets the BigGAN generator (Brock et al., 2019), which learns the categorical information through CCBN with ReLU activation. Investigating other architectures, like the style modulation layer (Karras et al., 2019) with Leaky ReLU activation (Xu et al., 2015), can be one of the future works. Besides, our approach mainly focuses on interpreting and utilizing a well-learned model for downstream tasks. How the insights provided in this work can inspire the design of a more powerful GAN model would be of more significance.

**Reproducibility Statement.** To ensure the reproducibility of our work, we provide the pseudo-code of the core implementation of our channels awareness in Appendix A. In addition, we provide implementation details of quantitative evaluation and synthesis performance evaluation in Sec. 4.1 and Appendix H, respectively. Implementation details of other analyses and applications can be found in the corresponding paragraph. We will also make our source code publicly available for all the core experiments.

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

## APPENDIX

## A  PSEUDO-CODE AND RUNNING TIME

Given a pre-trained class conditional model, our category-oriented channel awareness and latent-oriented channel awareness, can be computed *unsupervisedly and efficiently*. The below PyTorch-style (Paszke et al., 2019) pseudo-code provides the core implementation.

```
import torch

def compute_channel_awareness(ccbn_scale, ccbn_bias, e, num_samples, z_dim):
    """Computes the channel awareness with respect to a particular category.

    Args:
        ccbn_scale: A PyTorch module (usually a fully-connected layer) used
            in CCBN to compute the category-specific 'scale' coefficients.
        ccbn_bias: A PyTorch module (usually a fully-connected layer) used
            in CCBN to compute the category-specific 'bias' coefficients.
        e: The embedding vector for the target category.
        num_samples: Number of random samples used for computation.
        z_dim: Dimension of the latent space.

    Returns:
        A two-element tuple, indicating the category-oriented channel
            awareness and latent-oriented channel awareness, respectively.
    """
    # Random sample.
    z = torch.randn(num_samples, z_dim)
    e = e.reshape(1, -1).repeat(num_samples, 1)
    code = torch.cat([z, e], dim=1)

    # Channel probe, with shape [N, C].
    probe = - ccbn_bias(code) / ccbn_scale(code)
    # Category-oriented channel awareness, with shape [C].
    category_awareness = - torch.mean(probe, dim=0)
    # Latent-oriented channel awareness, with shape [C].
    latent_awareness = torch.var(probe, dim=0)

    return category_awareness, latent_awareness
```

**Running Time.** Computing channel awareness is *highly efficient*. With one NVIDIA Tesla V100 GPU, it costs $0.0017s$ to interpret all channels in the first layer of BigGAN regarding a certain class. For all layers, the average running time is $0.0012s$. The computing time for each layer is averaged using 200 classes.

## B  QUANTITATIVE EVALUATION ON BIGGAN-DEEP

This section provides additional evaluations for Sec. 4, including implementation details and results for evaluating the category-oriented channel awareness on the BigGAN-Deep model (Brock et al., 2019). Unlike BigGAN, BigGAN-Deep has twice the number of channels (*e.g.*, 2,048 channels in layer 0), making the single-channel evaluation more time-consuming. Besides, zeroing out a single channel makes an invisible change on the output image. Thus in this experiment, we zero out five channels for each intervention, and every five channels are selected along with the order of sorted channel indices by the category-oriented channel awareness. Results of top four results of four classes are shown in Fig. A1. Here we scale the two curves into the range of 0-1 for a better view.

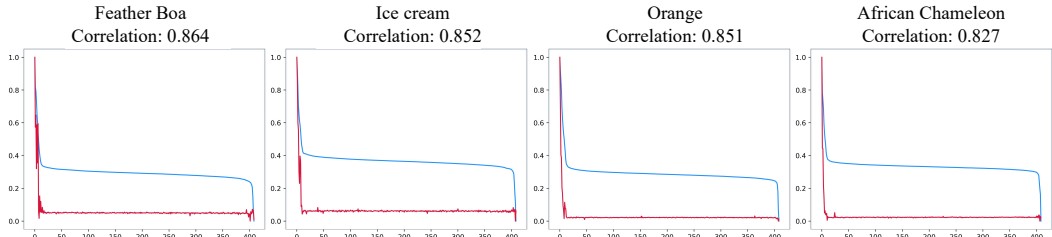

Figure A1: **Quantitative evaluation of category-oriented channel awareness on BigGAN-Deep** with five-channel modulation. The correlation between awareness score (blue) and categorical score change (red) appears on top of each figure. *X-axis:* intervention order following the sorted awareness score. *Y-axis:* Scaled values of both lines. The average correlation of all 1,000 classes is 0.603.

## C  MORE ANALYSIS OF LATENT-ORIENTED CHANNEL AWARENESS

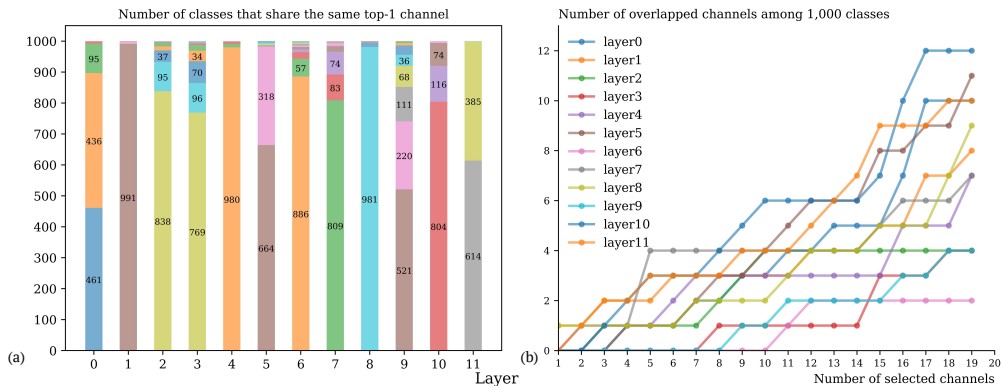

Figure A2: **In-depth analyses on the number of shared channels among all classes.** (a) We count how many classes share the top-1 channel at different layers, where the most popular channel at all layers serves over 450 classes from 1,000 classes in total. (b) Number of overlapped channels (X-axis) regarding different top-k values (Y-axis). Along with more channels incorporated for each class, the number of overlapped channels increase. (Served as additional results of Sec. 4.2 in the main paper.)

## D  MORE QUALITATIVE RESULTS OF CHANNEL MASKING

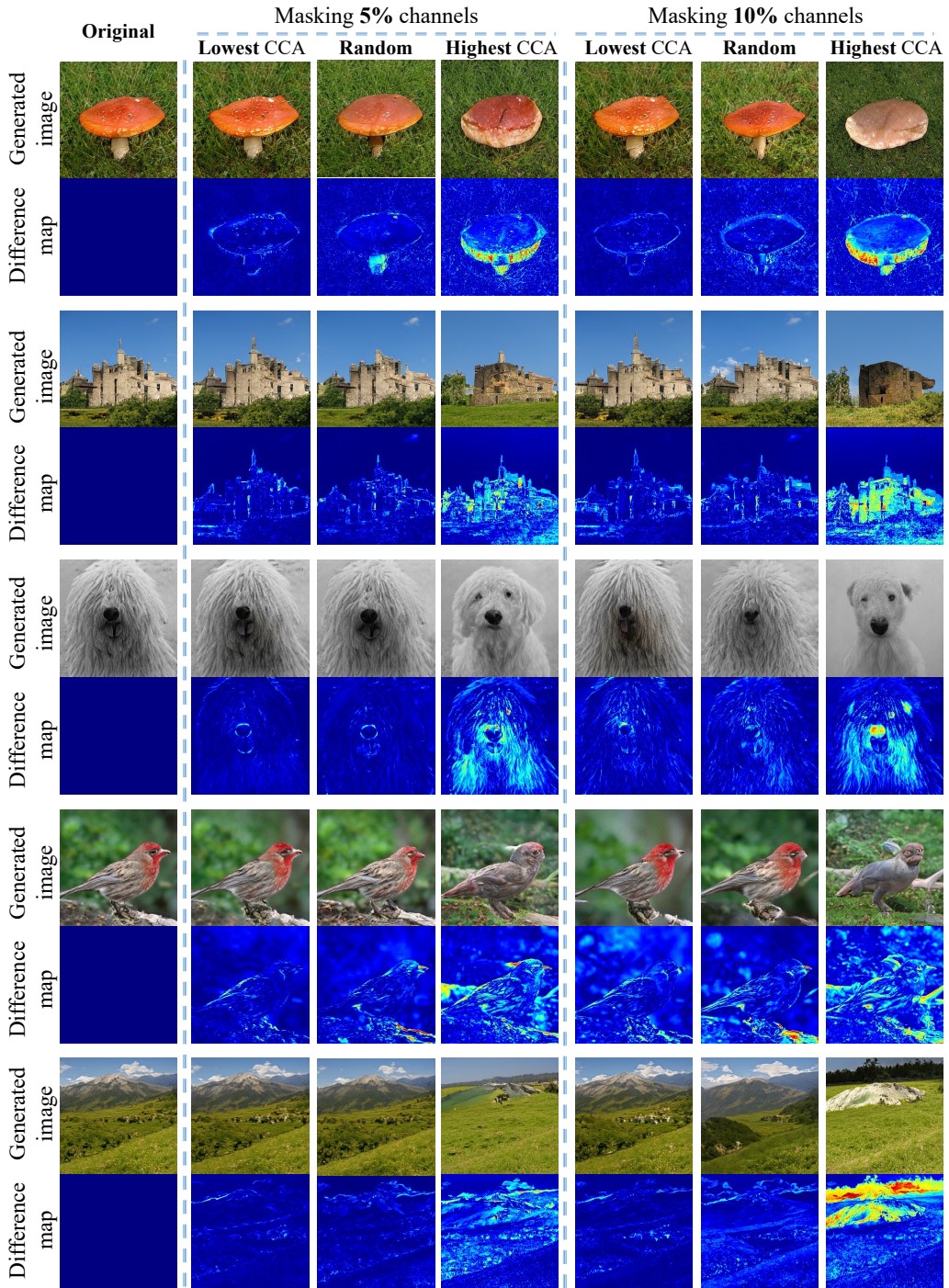

Figure A3: **More results of channel masking for quanlitative evaluation (Fig. 3 in the main paper),** which show comparisons among masking lowest (left) and highest (right) category-oriented channel awareness (CCA) as well as masking randomly selected channels (middle). The four class shown here are mushroom, castle, komondor, house finch, and alp.

# E    MORE RESULTS OF CATEGORY-ORIENTED ATTRIBUTE EDITING

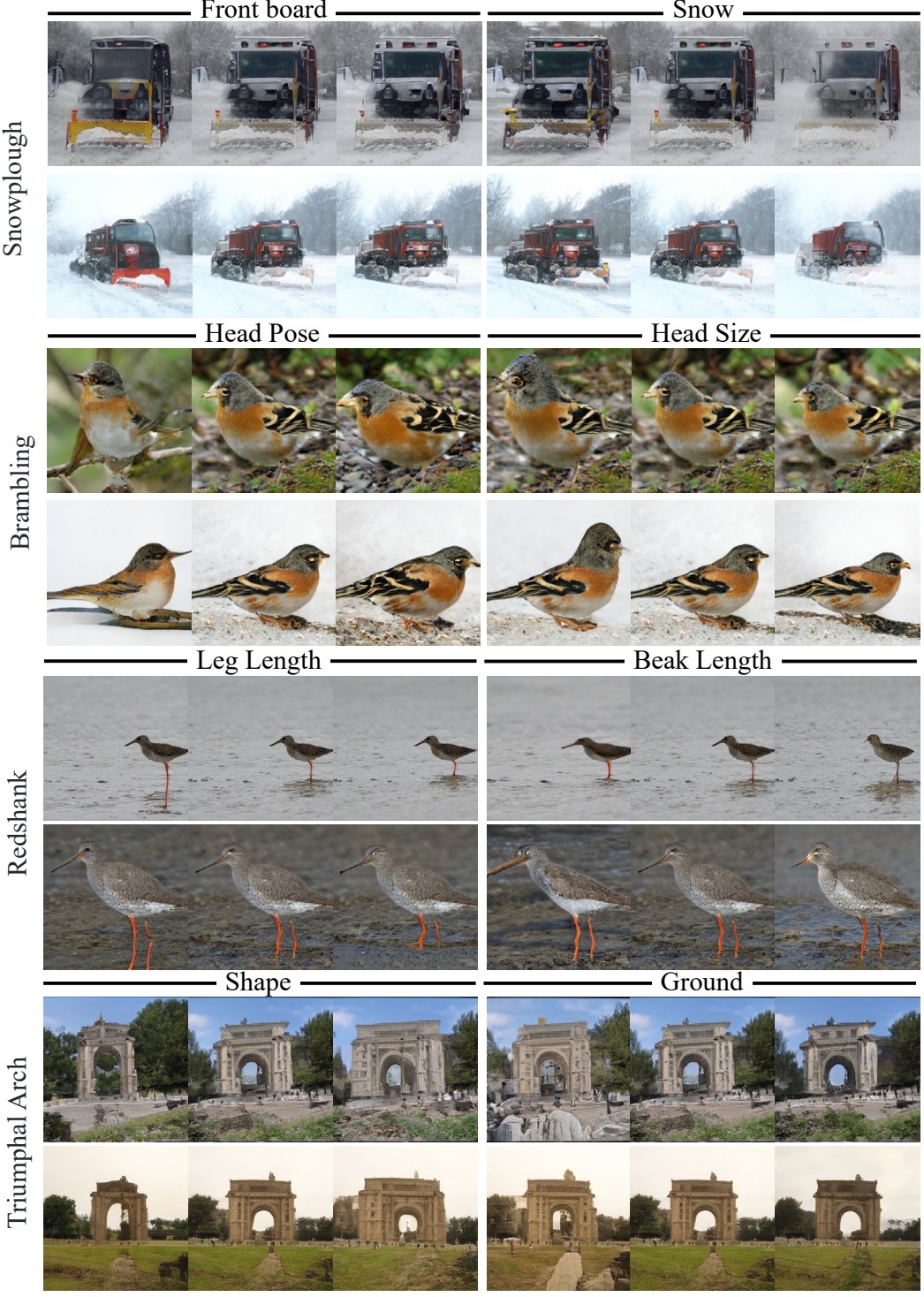

Figure A4: **Category-oriented attribute editing** (Sec. 5.1 in the main paper). Given a target class, our category-oriented channel awareness can unsupervisedly identify channels responsible for the class-oriented attributes, based on which image editing can be achieved via altering a single channel. For each group of images, the middle one is input, while the left and the right one are manipulating the single channel along the positive and the negative direction. Corresponding class names are annotated on the left.

# F MORE RESULTS OF LATENT-ORIENTED ATTRIBUTE EDITING

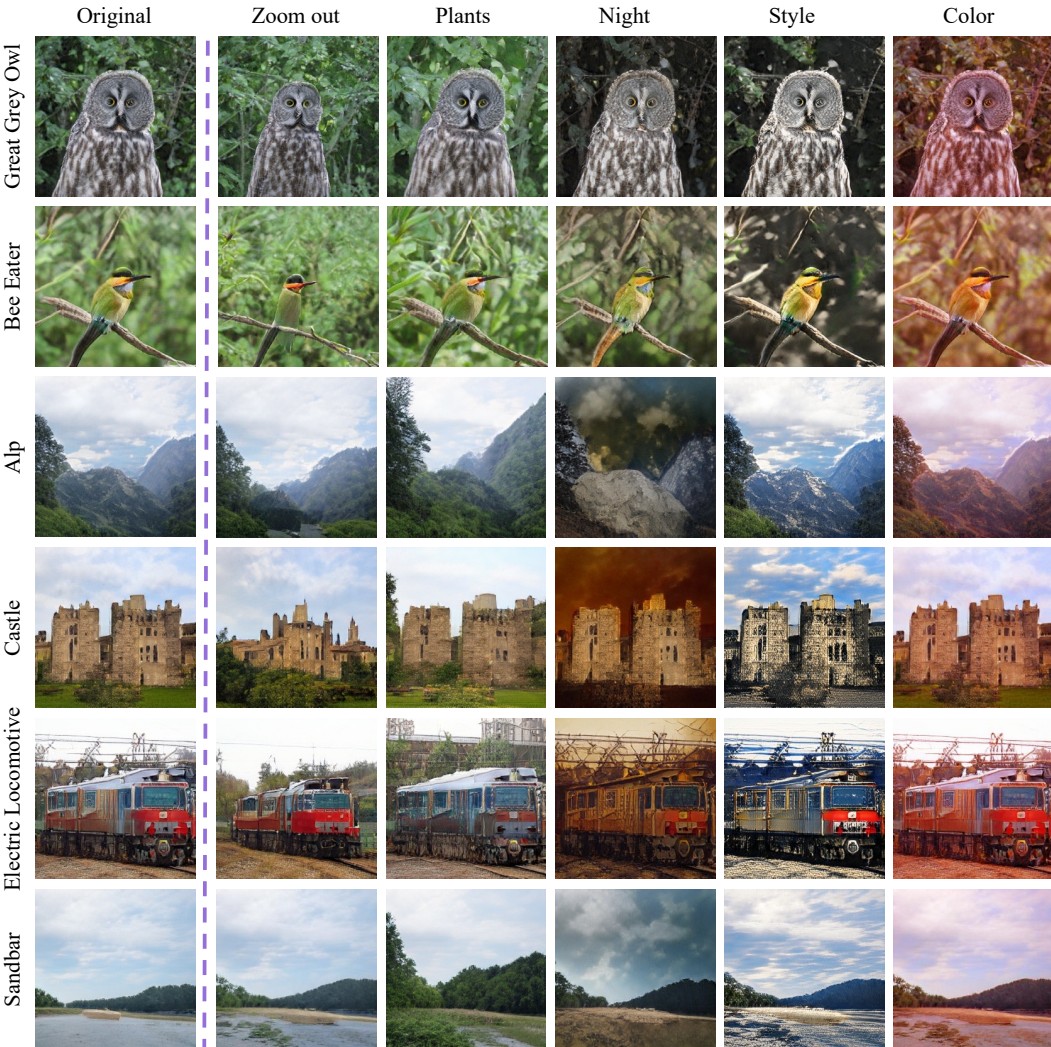

Figure A5: **Latent-oriented attribute editing** (Sec. 5.1 in the main paper). Our latent-oriented channel awareness unsupervisedly identifies channels that deliver information to all classes. Thus altering one channel can edit images from multiple classes. Corresponding class names are annotated on the left.

## G    MORE RESULTS OF CATEGORY HYBRIDIZATION

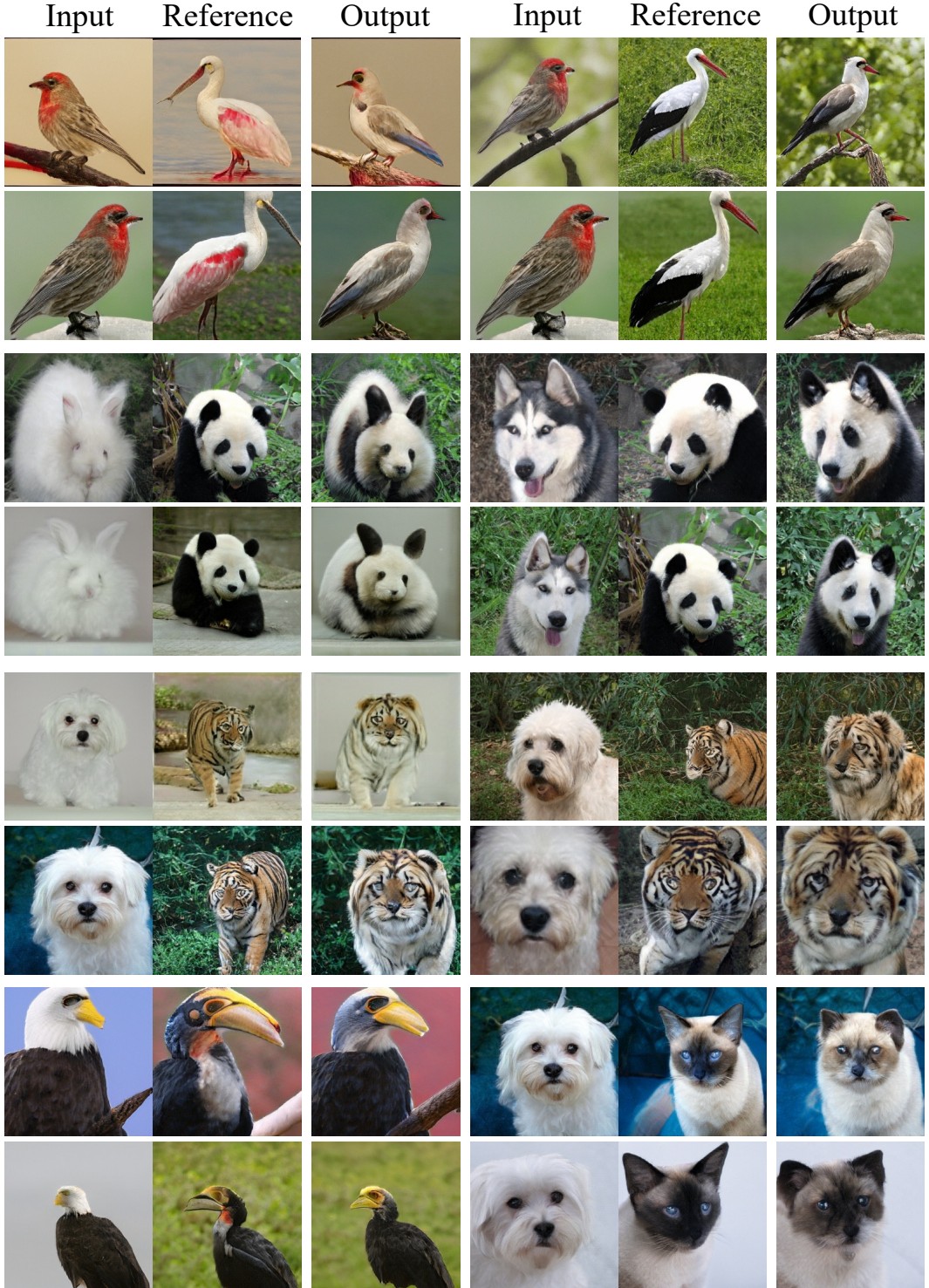

Figure A6: **Category hybridization** by mixing the channels that are relevant to two different categories (Sec. 5.2 in the main paper). For each group, the first two columns present the original syntheses, while the third column shows the hybridization result, which successfully fuses the characteristics (including both shape and appearance) of both classes.

# H    IMPLEMENTATION DETAILS OF SYNTHESIS PERFORMANCE EVALUATION

This section gives the implementation details for category-wise synthesis performance evaluation via total channel awareness (Sec. 5.4 in the main paper).

**Data preparation.** For precision, recall, and FID, which need to compare fake images with real images during estimation, we firstly pre-process real images from ImageNet (Deng et al., 2009) following the procedure in BigGAN training which is cropping and resizing to $256 \times 256$. The number of real images in ImageNet is around 1,300 for each class. We randomly sample all generated images without truncation for fairly measuring the synthesis performance.

**Precision and Recall.** Precision (Kynkäänniemi et al., 2019) is used for measuring sample quality by retrieving fake images and checking whether it locates within the manifold of real images. Recall (Kynkäänniemi et al., 2019) is used to measure sample diversity by retrieving real images and checking whether it is within the manifold formed by fake images. The ranges of these two metrics are both between zero to one. A higher value indicates a better performance. We calculate precision and recall with the number of fake images equal to the number of real images of each class followed the suggested setting (Kynkäänniemi et al., 2019). The feature extraction network is the pre-trained VGG-16. We set the neighborhood size $k$ equal to 3, which is a more robust choice(Kynkäänniemi et al., 2019).

**MS-SSIM.** We calculate the similarity between fake image pairs via the image similarity metrics MS-SSIM following (Odena et al., 2017). Lower image similarity indicates better diversity. For each class, we randomly sample 100 fake images to construct 10,000 image pairs for calculating similarity, then an averaged similarity score for the class can be obtained as the final result for the intra-class diversity.

**Fréchet Inception Distance (FID).** FID is a general metric that considers both diversity and quality by directly estimating the distance between the feature distribution of fake and real images. We calculate the FID with 50,000 generated images for each class. Then we exploit a pre-trained Inception-v3 as the feature extractor for estimating the feature distance between real and fake images.

