# OpenReview forum: "Interpreting Class Conditional GANs with Channel Awareness"
_ICLR.cc/2023/Conference — Submitted to ICLR 2023_

### Official Review · Reviewer_vDMD · 2022-10-21

**Confidence:** 4
**Correctness:** 3
**Technical Novelty And Significance:** 2
**Empirical Novelty And Significance:** 3
**Recommendation:** 5

**Clarity, Quality, Novelty And Reproducibility:**

#### Major
1. Generality: This work mainly relies on BigGAN. Since this work is based upon class-conditional (re)scaling + ReLU setting, which is quite general in class-conditional GAN, other pretrained models can be used for applying this technique. It would be great if the author could present the result with a completely different architecture and discuss the similar/different trends between those models.

2. Comparison with other methods: The author claimed that this approach differs from others, but I failed to see any of the baselines in the manuscript. For instance, I think (Wu et al., 2021) for (2) can be compared with this method when we provide a fixed set of classes embedding a specific class. For various experiments, it would be great if the author could present the difference between their approach and others for all experiments.


#### Minor
- Figure 1 caption (line 8): maybe you want to remove the period at the end of the sentence before the parenthesis?
- Figure 5 caption (line 2): "e select" -> "We select"
- Table 1 caption (line 5): "more ks" -> how about replacing all k there with $k$?
- Page 6 line 2: classes(e.g.,  -> classes (e.g.,

**Strength And Weaknesses:**

#### Strength
- This paper is well-written and easy to follow.
- Carefully drawn figures help the reader catch what this paper is about and the implications of their findings.
- Application in various ways, including segmentation examples, sounds interesting to explore.

#### Weakness
- Although a few methods are mentioned in related work, estimating the performance gap between this method and the rest of the literature is tough.
- Since this work is mainly based on BigGAN (+ although supplementary presents few details about BigGAN-Deep), it is hard to catch whether this method generally works for other class-conditional GANs beyond BigGAN.

**Summary Of The Paper:**

This work tries to understand what a particular generative adversarial network (GAN) learns and how we can use a pretrained GAN. Specifically, this model mainly analysis one of the class-conditional GANs named BigGAN with a channel probing technique. Although its findings and application look attractive at first, it is hard to estimate the true effect before comparing them with the previous work or other models that meet the class-conditioning + ReLU requirement.

**Summary Of The Review:**

This model tries to understand what class-conditional GAN learns and how we can play on top of the pretrained models. Although the results and the applications are fun to see, I would like to see a more direct comparison with other methods and other GAN models.

---

> ### Author Response · Authors · 2022-11-18
> **Response to Reviewer vDMD**
>
> Thank you very much for your useful and valuable suggestions.
>
> **A more direct comparison with other methods and other GAN models:**
>
> **Comparison with latent-space editing approaches regarding latent-oriented attributes editing.**
> We provide the comparison results on this anonymous link: https://ufile.io/eb69rlal.
>
> **Comparison with latent-space editing approaches regarding class-oriented attributes editing.**
> We provide the comparison results on this anonymous link: https://ufile.io/f0zy6zel.
> We applied a semantic discovery approach GANspace[2] devised for latent space on the class embedding space to find the class-related attributes. However, from the provided results, we can see that the vector directions in the class embedding space lack meaningful manipulations. That is why we further explore the channel space of class-conditional GANs.
>
> **Comparison with style mixing[1] regarding category hybridization.**
> See the third column of each group in Figure 7 of the main paper.
>
>
> [1] Karras, Tero, Samuli Laine, and Timo Aila. "A style-based generator architecture for generative adversarial networks." Proceedings of the IEEE/CVF conference on computer vision and pattern recognition. 2019.
> [2]  Härkönen, Erik, et al. "Ganspace: Discovering interpretable gan controls." Advances in Neural Information Processing Systems 33 (2020): 9841-9850.

---

### Official Review · Reviewer_EESf · 2022-10-24

**Confidence:** 4
**Correctness:** 3
**Technical Novelty And Significance:** 2
**Empirical Novelty And Significance:** 3
**Recommendation:** 5

**Clarity, Quality, Novelty And Reproducibility:**

This paper is overall well-written and it is easy to follow. There are some missing details and references that should be provided in the paper. Please find more details in *Strength And Weaknesses.

**Strength And Weaknesses:**

### Strengths
This paper is well-written and it is easy to follow.
The application of category hybridization, segmentation and synthesis evaluation is interesting.
The authors promise open-source code.


### Weaknesses

**Clarity**

There are many missing details. For example,
1. the authors forget to introduce the images that appeared in Figure 1(b). What do they mean? What are the differences between red, green, and blue bounding boxes?
2. The authors labeled the edited images in Figure 6. Are they labeled manually or by pre-trained classifiers?
3. The authors should provide more details about how to achieve category hybridization and image editing in Section 5.
4. In Figure 7, the authors mixed similar categories, e.g., monkey and dog are both animals, How about mixing the monkey and a car? Or mixing a tree and a dog?

**Novelty**

1. There are some missing references that have studied the relationships between the channels and the categories (i.e., semantics), and they also achieved image editing by editing channels [A]. The authors should discuss and compare with them.

[A] Learning semantic-aware normalization for generative adversarial networks. H Zheng, J Fu, Y Zeng, J Luo, ZJ Zha. Advances in Neural Information Processing Systems 33, 21853-21864.

**Quality**

1. The authors study channel awareness only on one model (i.e., BigGAN) on one dataset (i.e., ImageNet), which limits the credibility of channel awareness. We suggest the authors study it on more GAN models and more datasets to verify it.
2. In Section 3, the authors calculate category-oriented channel awareness on sampled images. How many samples? 1,000 or 10,000 samples mentioned in the implementation details are not convincing enough since ImageNet has 14 million images.
3. In Figure 7, the authors claimed they perform better style mixing than StyleGAN, however, they didn’t provide the results of StyleGAN for comparisons.


**Summary Of The Paper:**

This paper proposes channel awareness based on class-conditional batch normalization (CCBN) to study how a single channel contributes to the final synthesis. Specifically, they conduct experiments on BigGAN pre-trained on ImageNet. The authors show that only a subset of channels primarily contributes to a specific category and some channels are shared by similar categories or all classes. Finally, they show the applications by editing specific channels for image editing, category hybridization, segmentation and image synthesis evaluation.


**Summary Of The Review:**

The presentation of this paper is clear. My concerns are mainly about the novelty and the quality mentioned in *Strength And Weaknesses. Specifically, the authors conduct experiments only on BigGAN on ImageNet, which is not convincing enough. The authors should also provide comparisons with some SOTA approaches (e.g., StyleGAN). If all the concerns mentioned in Weaknesses are addressed, I will raise my rating.

---

> ### Author Response · Authors · 2022-11-18
> **Response to Reviewer EESf**
>
> Thank you very much for your useful and valuable suggestions.
>
> **Clarity**
> **Meanings of images and differences between bounding boxes that appeared in Figure 1(b).**
> Images in Figure 1(b) stand for visualizations of different classes. The yellow circle with many class images indicates those channels are shared among all classes. Bounding boxes with three different colors are used to match the color of the category-specific channels plotted below those categories for a better view. We draw this figure carefully in order to demonstrate our finding that different categories own different sets of channels, and some channels are shared among all classes. We will revise the description of this figure to make it clearer.
>
> **Labels of the edited images in Figure 6.**
> Labels of edited images are labeled manually.
>
> **How to achieve image editing and category hybridization in Section 5.**
> **Image Editing**
> We edit generated images by multiplying or adding a constant on a single channel during forwarding the generator. The constant is usually three or five as manipulation strength. Multiplying and adding on the same channel result in similar editing results.
> We edit only the top 5 channels in each layer with the highest category-oriented channel awareness (for category-oriented attributes) or latent-oriented channel awareness (for latent-oriented attributes) and showcase partial editing results due to the limited space.
> We can provide the specific channel id for reproducing our editing results.
>
> **Category Hybridization**
> We achieve category hybridization of two generated samples by transplanting multiple channel maps responsible for the reference class to the target class during forwarding the generator.
> Also, the channels responsible for the reference class are located by category-oriented channel awareness with the highest scores. In practice, the number of transplanted channels adjusts the hybridization strength: transplanting more channels from the reference category, the hybridization results will be more like the reference category.
>
> **Novelty**
> **Discussion and comparison with [A].**
> Our work aims to interpret off-the-shelf class-conditional GAN models to answer the question of how class-conditional GANs (i.e., BigGAN) manage to synthesize one thousand various categories in one network. However, [A] still works for unconditional image generation, which is single-class networks (such as faces, cats, or cars). [A] also shows results on a conditional image inpainting task. However, this is also far away from our objective: understanding the generation mechanism of class-conditional GANs: How it unifies the synthesis of one thousand distinct categories.
>
> **Quality**
> **The number of calculated samples is not enough since ImageNet has 14 million images.**
> As mentioned in section 5 of the original paper of BigGAN[1], they use ImageNet ILSVRC 2012[2] dataset for their models. From the website of ImageNet (https://image-net.org/download.php), we can check that the dataset for the visual recognition challenge contains 1,281,167 training images which are around one million in total and are around one thousand images per category. Hence, we calculate category-oriented channel awareness and categorical score change on 10,000 and 1,000 samples per category, which is enough since the number of training images per category is only 1,000.
>
> **Comparison with style mixing proposed in StyleGAN.**
> In fact, we provide the results of style mixing in the third column of each group of images in Figure 7, which we have mentioned in its caption. We apologize for the confusion, and we will revise this part and highlight this information in Figure 7.
>
> [1] Brock, Andrew, Jeff Donahue, and Karen Simonyan. "Large scale GAN training for high fidelity natural image synthesis." arXiv preprint arXiv:1809.11096 (2018).
> [2] Russakovsky, Olga, et al. "Imagenet large scale visual recognition challenge." International journal of computer vision 115.3 (2015): 211-252.

---

### Official Review · Reviewer_dhhh · 2022-10-24

**Confidence:** 4
**Correctness:** 4
**Technical Novelty And Significance:** 2
**Empirical Novelty And Significance:** 2
**Recommendation:** 5

**Clarity, Quality, Novelty And Reproducibility:**

The work appears to be original, it is well written and easy to follow. The novelty, and potentially the applicability of the paper's ideas are limited (see weaknesses above).

**Strength And Weaknesses:**

Strengths
---

The paper presents a very simple method for quantifying the impact of each feature channel in the generation process of a BigGAN (by studying its conditioning mechanism, i.e., CCBN). Its simplicity is its main strength, while the provided applications are useful and potentially useful.



Weaknesses
---

My main point of criticism concerns the generality of the proposed method in conditional GANs that do not use CCBN as a conditioning mechanism. The paper investigates the use of CCNB in BigGANs, but leaves without any discussion another recent and powerful conditional GAN, namely StyleGAN-XL [1]. To the best of my knowledge, BigGAN is the only relatively recent conditional generator to use CCBN as a conditioning mechanism. This renders the paper BigGAN-specific, and potentially reduces its applicability in other class conditional architectures, mainly StyleGAN-XL that gives significantly better results, as far as I know.

Moreover, another weakness of the paper concerns its novelty in the reported applications. The paper states that the proposed method enables several novel applications (such as image editing and segmentation) with conditional GANs, but this is not accurate. Conditional GANs have been studied in terms of finding interpretable directions (e.g., [2,3]), as well as in the applications of image editing [2,3,4] and segmanation [3,5].

Finally, whilst the above applications do seem interesting and the provided method helps towards, the "key findings" of the paper, as stated in the manuscript (and listed above), are not surprising or totally new.


References

[1] Sauer, Axel, Katja Schwarz, and Andreas Geiger. "StyleGAN-XL: Scaling stylegan to large diverse datasets." ACM SIGGRAPH 2022 Conference Proceedings. 2022.
[2] Tzelepis, Christos, et al.. "WarpedGANSpace: Finding non-linear RBF paths in GAN latent space." Proceedings of the IEEE/CVF International Conference on Computer Vision. 2021.
[3] Voynov, Andrey, and Artem Babenko. "Unsupervised discovery of interpretable directions in the gan latent space." International conference on machine learning. PMLR, 2020.
[4] Oldfield, James, et al. "Tensor Component Analysis for Interpreting the Latent Space of GANs." arXiv preprint arXiv:2111.11736 (2021).
[5] Andrey Voynov, et al. Big GANs are watching you: Towards unsupervised object segmentation with off-the-shelf generative models. arXiv preprint arXiv:2006.04988, 2020

**Summary Of The Paper:**

This paper investigates the conditioning mechanism of BigGANs, i.e., Class-Conditional Batch Normalization (CCBN), and proposes "channel awareness" in order to quantify the impact of each feature channel in the the final synthesis. Using channel awareness, the paper asserts that a) only a subset of channels is primarily responsible for the generation of a particular category, b) that similar categories (such as cats and dogs) share some channels (they are activated jointly), and c) that some channels are "global" in the generation  process of all classes. Finally, the paper presents results on a number of downstream tasks (image editing, hybridization between two classes, image segmentation).

**Summary Of The Review:**

The main reason I cannot suggest acceptance of the paper at this time is the limited applicability of the proposed method, since it only studies CCBN (BigGAN), but not other class-conditional generators, mainly StyleGAN-XL.

---

> ### Author Response · Authors · 2022-11-18
> **Response to Reviewer dhhh**
>
> Thank you very much for your useful and valuable suggestions.
>
>
> **Novelty of applications**
> **Image editing**
> Our work has two clear differences compared with the mentioned related work:
> (1) Latent space and channel space: [2,3,4] achieve image editing by manipulating the global latent vectors in latent space, which simultaneously alter all channels. Differently, we study the channel space, and we discover that simply altering **a single** class-aware channel or latent-aware channel can also achieve semantically meaningful image manipulations.
> To the best of our knowledge, we are the first to study the relationship between channel and class in class-conditional GANs.
> (2) Approach: Our approach does not require training any extra network in [2, 3] and an optimization process in [4] to achieve image editing. We achieve this by simply multiplying or adding a constant on the identified class-aware channel or latent-aware channel, which is totally different from prior works.
> We compare our approach with several latent space approaches in anonymous links. Please note that discovering latent vectors for semantic manipulation can only find latent-related attributes (see https://ufile.io/eb69rlal). Applying these approaches to class embedding space fails to discover meaningful directions for manipulating class-related attributes (see https://ufile.io/f0zy6zel).
>
> **Conditional GANs for image segmentation**
> For image segmentation, we achieve a more fine-grained segmentation than [5]. Specifically, [5] mainly works for object-background segmentation. We achieve more detailed segmentations, including object parts such as organs for animals and components for cars.
> In addition, we achieve finer results via a very simple K-means clustering algorithm. This is because our findings of channel awareness help to highlight the most important channels containing important object parts information.
> Hence, despite the task of utilizing pretrained GANs for image editing and segmentation that has been explored in previous work, we achieve semantic editings and finer segmentation via a surprisingly simple and novel approach because of our findings of channel awareness.
> Thank you very much for pointing out these concerns. We will revise the paper to make the description of applications and our approach to these applications more accurate.

---

### Author Response · Authors · 2022-11-18
**General Response**


We thank all the reviewers for their valuable comments. We are encouraged that they find our approach and applications **useful** (dhhh), **interesting** (EESf, vDMD), **simple and effective** (dhhh).

The main concerns lie in the number of class-conditional GANs and datasets for experiments. Before addressing each individual issue, we would like to **reaffirm the significance of this work:**
(i) Compared with unconditional GANs, conditional GANs unify the generation of multiple categories in a super-efficient manner yet remain less explored. This work takes the first step and unlocks the great potential of conditional GANs with four applications.
We hope that our discovery could inspire more thoughtful ideas.
(ii) This work manages to quantify the contribution of a single channel to the final synthesis in conditional GANs. Through interpreting such a generation mechanism, more studies of improving GAN training could be expected.

**Concerns about experiments on only one model and one dataset.**
Besides BigGAN, we also evaluate our method on BigGAN-deep (see supplementary material). We choose the BigGAN family as it is the most representative conditional generative model based on GANs and has a good performance. We choose ImageNet dataset considering its complexity and large diversity, making it more representative and convincing than the toy models trained on CIFAR-10 or MNIST. We have also discussed the limitation of our approach (Section 6) since it highly relies on the CCBN module, but still it fills in the gap of interpreting conditional GANs.

---

### Decision · Program_Chairs · 2023-01-20

**Decision:**

Reject

**Justification For Why Not Higher Score:**

All reviewers lean toward reject.  After the author response, Reviewer vDMD comments about specific concerns that are still unresolved.  The AC concurs.

**Justification For Why Not Lower Score:**

N/A

**Metareview: Summary, Strengths And Weaknesses:**

This paper proposes techniques for manipulating or editing GAN output through the behavior of feature channels within GANs that utilize class-conditional batch normalization (CCBM).  Some concerns are each raised by multiple reviewers: (1) limited generality of the method due to reliance on CCBM; how would it apply to other GAN architectures? (2) novelty with respect to reported applications; (3) experimental comparison to prior work on controllable generation using GANs.  While the authors provide a rebuttal, it did not move reviewers to update scores.  Reviewer vDMD commented that the response did not resolve concerns about generality and experimental comparison to baselines.  The Area Chair agrees with this assessment and does not see a basis for accepting the paper.